

# External validation of cut-off points for foveal thickness taking into account the intraretinal fluid using optical coherence tomography to diagnose diabetic macular oedema

Carmen Hernández-Martínez[1], Antonio Palazón-Bru[2], Cesar Azrak[1], Aída Navarro-Navarro[1], Manuel Vicente Baeza-Díaz[1], José Juan Martínez-Toldos[1] and Vicente Francisco Gil-Guillén[2,3]

[1] Ophthalmology Service, General Hospital of Elche, Elche, Alicante, Spain
[2] Department of Clinical Medicine, Miguel Hernández University, San Juan de Alicante, Alicante, Spain
[3] Research Unit, Elda Hospital, Elda, Alicante, Spain

Corresponding author
Antonio Palazón-Bru,
antonio.pb23@gmail.com

## ABSTRACT

**Background.** In late 2015, cut-off points were published for foveal thickness to diagnose diabetic macular oedema taking into account the presence of intraretinal fluid using optical coherence tomography (OCT) in primary care patients (90 μm in the presence of intraretinal fluid and 310 μm otherwise).

**Methods.** This cross-sectional observational study was carried out on 134 eyes of diabetic patients treated in specialised ophthalmology services in a Spanish region in 2012–2013, to externally validate the aforementioned cut-off points. The main variable (Clinical Standard) was the diagnosis of macular oedema through indirect ophthalmoscopy and posterior segment slit-lamp biomicroscopy. As validation variables, both the foveal thickness and the presence of intraretinal fluid obtained by OCT were used. Validation was performed using bootstrapping by calculating the area under the ROC curve (AUC), sensitivity, specificity, positive likelihood ratio (PLR) and negative likelihood ratio (NLR).

**Results.** Forty-one eyes presented diabetic macular oedema (30.6%). The bootstrapping validation parameters were: AUC, 0.88; sensitivity, 0.75; specificity, 0.95; PLR, 14.31; NLR, 0.26. These values were very similar to those of the original publication.

**Conclusion.** We have externally validated in specialised care patients the cut-off points published for the diagnosis of diabetic macular oedema. We suggest that others carry out validation studies in their communities.

# INTRODUCTION

Diabetic macular oedema is the leading cause of poor vision in diabetic patients (*Klein et al., 1989*). For diagnosis, the reference method (Clinical Standard) is posterior segment

biomicroscopy, the stereoscopic examination of the fundus of the eye using a magnifying lens with or without contact with the pupil in mydriasis (*Anonymous, 1985*; *Kinyoun et al., 1989*). Screening for diabetic retinopathy or diabetic macular oedema in primary care centres is performed using non-stereoscopic retinography (photographic colour images to control the progression of fundus of the eye abnormalities). However, diffuse thickening or thickening in the form of retinal cysts as the initial sign of diabetic macular oedema may go unnoticed using this method, as it only allows the detection of macular oedema in the presence of hard exudates, macular haemorrhage or microaneurysms (*Gómez-Ulla et al., 2002*; *Baeza Diaz et al., 2004*; *Baeza et al., 2009*).

Optical coherence tomography (OCT), which has been available for several years, allows us to perform a quantitative and qualitative study of diabetic macular oedema by providing a cross-sectional image of the retina, at high magnification, and automatically measures the thickness of various retinal layers. The OCT also shows the abnormalities in structures such as the presence of cysts and the accumulation of fluid (*Hee et al., 1995*; *Hee et al., 1998*; *Browning et al., 2004*). Although OCT gives us two parameters to detect diabetic macular oedema, the literature has focused mainly on foveal thickness (*Virgili et al., 2015*), trying to determine a cut-off point from which we can say that there is diabetic macular oedema. In other words, the influence of intraretinal fluid in the macula in the OCT has not been considered, as when evaluating a diagnostic test it is possible that the presence of intraretinal fluid on OCT does not agree with the Gold standard defined according to ETDRS in 1985, which has been used for all the publications regarding OCT (*Anonymous, 1985*; *Virgili et al., 2015*). Thus, we could have patients without diabetic macular oedema as diagnosed by the Gold standard but in whom an OCT indicates the presence of intraretinal fluid. At the end of 2015, a paper was published that did combine both parameters (*Hernández-Martínez et al., 2015*). This study determined cut-off points for foveal thickness depending on whether OCT imaging revealed intraretinal fluid. In the presence of intraretinal fluid, the diagnostic test for diabetic macular oedema was considered positive when the patient had a foveal thickness value equal to or greater than 90 μm. Otherwise this value had to be equal to or greater than 310 μm. This latter value may seem high, in light of the cut-off points normally used in the diagnosis of diabetic macular oedema (*Virgili et al., 2015*). However, as mentioned above, the previous studies did not contemplate the joint analysis of foveal thickness plus the presence of intraretinal fluid on OCT. This should be analysed in depth before comparing the value 310 μm with the already known value. On the other hand, these cut-off points were internally validated through bootstrapping, obtaining both the area under the Receiver Operating Characteristic curve (AUC) and the calibration (sensitivity and specificity) (*Hernández-Martínez et al., 2015*). To date, these cut-off points have not been externally validated, that is, in a different sample of patients. Given the importance of the validation of a diagnostic test before its use, we performed this process in a sample of diabetic patients referred to ophthalmology services. In this validation we aimed to determine the sensitivity, specificity and AUC. If satisfactory values are found, we can then say that the test has been validated and may be implemented in clinical practice in our community, as we would have a more exact screening test for diabetic macular oedema. This would result in a reduction in the care load of ophthalmological services since the test

could be undertaken by health care personnel not specialised in ophthalmology with a latest generation OCT that focuses on the retina automatically (*Hernández-Martínez et al., 2015*).

## METHODS

### Study population

The study population comprised diabetic patients who were referred from primary health care centres to the ophthalmology services of the General University Hospital of Elche. Patients are primarily referred when they present a visual acuity deficit or there is suspected diabetic retinopathy or diabetic macular oedema in the screening tests (*Generalitat Valenciana: Conselleria de Sanitat, 2006*). This hospital has a catchment population of 169,555 inhabitants in the region of Bajo Vinalopó, which is located in the province of Alicante (southeast Spain). The health system is free and universal, both at the primary and specialised care levels.

### Study design and participants

The study design was cross-sectional observational with the objective of externally validating the diagnostic test published by *Hernández-Martínez et al. (2015)* in our geographical area. A random sample of diabetic patients who attended hospital ophthalmology services between October 2012 and June 2013 was selected. A random day was selected each week and all patients who attended that day and wanted to participate in the study were recruited. Patients were excluded if they had any of the following (*Azrak et al., 2015a*; *Azrak et al., 2015b*): cataract surgery in the last three months, laser treatment in the macular area or panphotocoagulation, dementia, treatment with intravitreal anti-angiogenic drugs, vitreoretinal surgery, high myopia or other macular conditions. The justification for these criteria is that they are conditions which can alter the anatomy of the macula.

### Variables and measurements

The main study variable (Clinical Standard) was the diagnosis of macular oedema through indirect ophthalmoscopy with a 28-diopter lens (28D aspheric lens, Volk Optical Incorporated, Mentor, OH, USA) and the stereoscopic exploration of the fundus with a Topcon SL-8Z Slit Lamp (Topcon Corporation, Tokyo, Japan) and a 78-diopter magnifying lens (Volk Optical Incorporated, Mentor, OH, USA) with the pupil in mydriasis. These examinations were carried out by an expert retinal ophthalmologist (*Anonymous, 1985*; *Kinyoun et al., 1989*). Through the analysis of the images obtained using these techniques, the ophthalmologist was able to confirm the presence of diabetic macular oedema, according to the following definition: the presence of hard exudates or retinal thickening located 500 µm from the fovea (*Azrak et al., 2015a*; *Azrak et al., 2015b*).

As validation variables, both the foveal thickness and the presence of intraretinal fluid were used (Fig. 1). These parameters were obtained through OCT (Topcon 3D OCT-2000; Topcon Corporation, Tokyo, Japan) and interpreted by a different expert retinal ophthalmologist. The foveal thickness is given automatically by the device software, which obtains images of the macular area through 512 horizontal and 128 vertical line scans, which focus on the fixation point, resulting in a 3D 6 mm × 6 mm volume cube

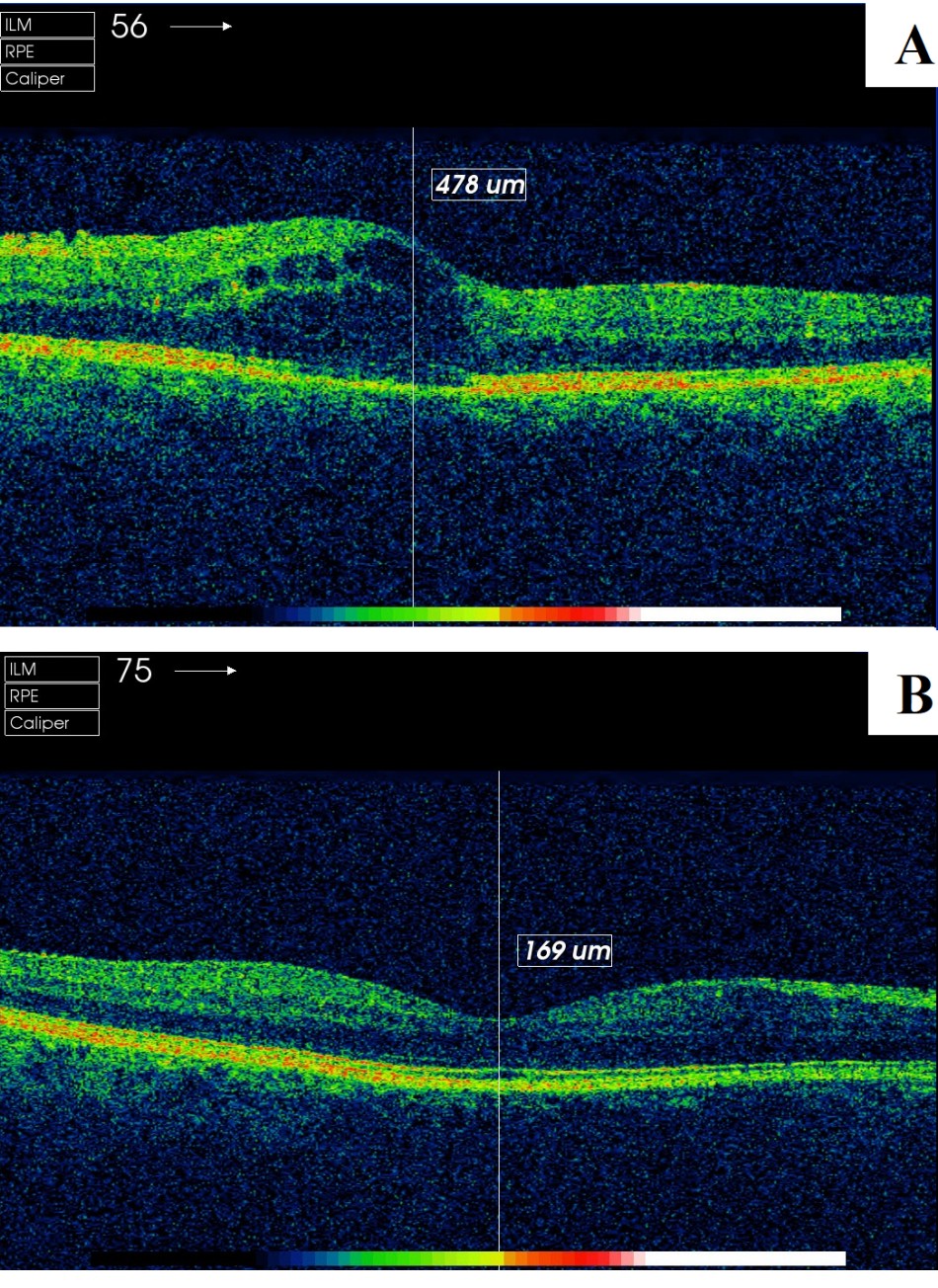

**Figure 1** **Screenshot of the retinal map analysis using Topcon 3D OCT-2000 (Topcon Corporation, Tokyo, Japan).** (A) patient with presence of intraretinal fluid; (B) patient without presence of intraretinal fluid. The copyright holder (Topcon Corporation®) has approved the utilization of this figure.

pattern. To perform the indicated measurement, an area of 6 mm in diameter was used, where the centre was the fovea, to thus obtain an area of 1,000 central microns for the assessment. To measure the presence of intraretinal fluid (cysts or hyporeflective spaces), the ophthalmologist examined the horizontal OCT image of the retina (B-Scan) (*Azrak et al., 2015a*; *Azrak et al., 2015b*; *Hernández-Martínez et al., 2015*).

At the descriptive level, to compare our population with that of the original study determining cut-off points (*Hernández-Martínez et al., 2015*), we used the electronic medical history, ophthalmological assessment and clinical interview with the patient, recording gender, type 2 diabetes mellitus, arterial hypertension, dyslipidaemia, smoking, age, HbA1c (%), years since diabetes diagnosis and visual acuity.

## Sample size calculation

Given that data were collected over a specific period of time, the sample size was calculated a posteriori, that is, determining whether the collected sample was sufficient to answer our research question (to validate a diagnostic test by calculating the sensitivity, specificity and AUC). Our sample size was 134 eyes. To estimate whether this sample was sufficient to validate the diagnostic test, we computed the specificity estimation using the following parameters: expected specificity value of 89% (*Hernández-Martínez et al., 2015*), prevalence of macular oedema in referred patients of 30.6% (*Azrak et al., 2015a*; *Azrak et al., 2015b*) and Type I error of 5%. With these parameters, a precision of 6.36% was obtained, which was satisfactory for our objective.

## Statistical methods

The quantitative variables were described by calculating the mean and standard deviation, and the qualitative variables with the absolute and relative frequencies. The descriptive characteristics of our population were compared with those of *Hernández-Martínez et al. (2015)*, through tests based on the $\chi^2$ distribution (Pearson or Fisher) and the Student's *t*-test. To determine the discrimination of the diagnostic test, the AUC was calculated using the formula proposed by Hernández-Martínez for the probability of diabetic macular oedema (Supplemental Material by *Hernández-Martínez et al. (2015)*). Sensitivity, specificity and positive likelihood ratio (PLR) and negative likelihood ratio (NLR) were calculated to measure the calibration (observed similar to expected). All these calculations were performed by bootstrapping (1,000 samples). A bootstrap sample is a random sample with replacement taken from the original sample and with the same number of elements as this. In each of these samples a statistical estimate is made, which allows its distribution to be constructed as 1,000 values are available for this distribution. This technique is the most accurate when validating predictive models (*Steyerberg et al., 2001*). All analyses were performed with a significance of 5% and the associated confidence interval (CI) was obtained for each relevant parameter. The statistical packages used were IBM SPSS Statistics 19 y R 2.13.2.

## Ethical issues

All patients gave their written informed consent. The study was approved by the Ethics Committee of the General University Hospital of Elche and complies with the World Medical Association Declaration of Helsinki and with the standards described by the European Union Guidelines for Good Clinical Practice.

**Table 1 Descriptive and comparative characteristics of our sample and the data by *Hernández-Martínez et al. (2015)*.**

| Variable | Our sample $n = 134$ n(%)/x ± sd | *Hernández-Martínez et al. (2015)* $n = 277$ n(%)/x ± sd | *p*-value |
|---|---|---|---|
| Diabetic macular oedema | 41(30.6) | 37(13.4) | <0.001 |
| Foveal thickness (μm) | 268.6 ± 79.5 | 270.4 ± 45.1 | 0.808 |
| Intraretinal fluid | 36(26.9) | 48(17.3) | 0.034 |
| Male gender | 66(49.3) | 162(58.5) | 0.097 |
| Type 2 diabetes mellitus | 109(81.3) | 249(89.9) | 0.023 |
| Hypertension | 85(63.4) | 137(49.5) | 0.010 |
| Dyslipidaemia | 63(47.0) | 141(50.9) | 0.526 |
| Smoking | 25(18.7) | 49(17.7) | 0.918 |
| Age (years) | 62.9 ± 15.0 | 61.6 ± 13.0 | 0.391 |
| HbA1c (%) | 7.7 ± 1.6 | 7.5 ± 1.7 | 0.255 |
| Years since diabetes diagnosis | 13.8 ± 9.9 | 14.1 ± 10.8 | 0.786 |
| Visual acuity | 0.7 ± 0.3 | 0.9 ± 0.2 | <0.001 |

**Notes.**

n(%), absolute frequency (relative frequency); x ± sd, mean ± standard deviation.

## RESULTS

The sample size was 134 eyes, of which 41 had diabetic macular oedema (30.6%, 95% CI [22.8–38.4%]); 36 were clinically significant macular oedema (26.9%, 95% CI [19.4–34.4%]) and 5 clinically non-significant macular oedema (3.7%, 95% CI [0.5–6.9%]). The descriptive characteristics of the sample analysed, as well as the comparison with the original population used in the design of the diagnostic test, are shown in Table 1. In this study, we observed the following significant differences ($p < 0.05$): higher prevalence of diabetic macular oedema in our population (30.6 versus 13.4%), higher presence of intraretinal fluid (26.9 versus 17.3%), lower prevalence of type 2 diabetes mellitus (81.3 versus 89.9%), higher prevalence of arterial hypertension (63.4 versus 49.5%), and lower mean visual acuity (0.7 versus 0.9).

The parameters of the bootstrapping validation are shown in Figs. 2–6. Regarding discrimination, we observed that the AUC had a mean value of 0.88 ± 0.04 (Fig. 2), indicating that it has a high discriminating power. Furthermore, in the calibration we obtained a mean sensitivity of 0.75 ± 0.07 (Fig. 3), specificity of 0.95 ± 0.02 (Fig. 4), PLR of 14.31 (interquartile range of 8.9) (Fig. 5) and NLR of 0.26 ± 0.07 (Fig. 6). All these indicate that the diagnostic test has been validated, since these results are very similar to those of the original work (*Hernández-Martínez et al., 2015*). Finally, the number of false positives per 1,000 referrals was 37.3 (95% CI [5.2–69.4]) and for false negatives it was 74.6 per 1,000 referrals (95% CI [30.1–119.1]).

## DISCUSSION

### Summary

This study externally validated the cut-off points for the diagnosis of diabetic macular oedema considering the presence of intraretinal fluid (90 μm when there is fluid and

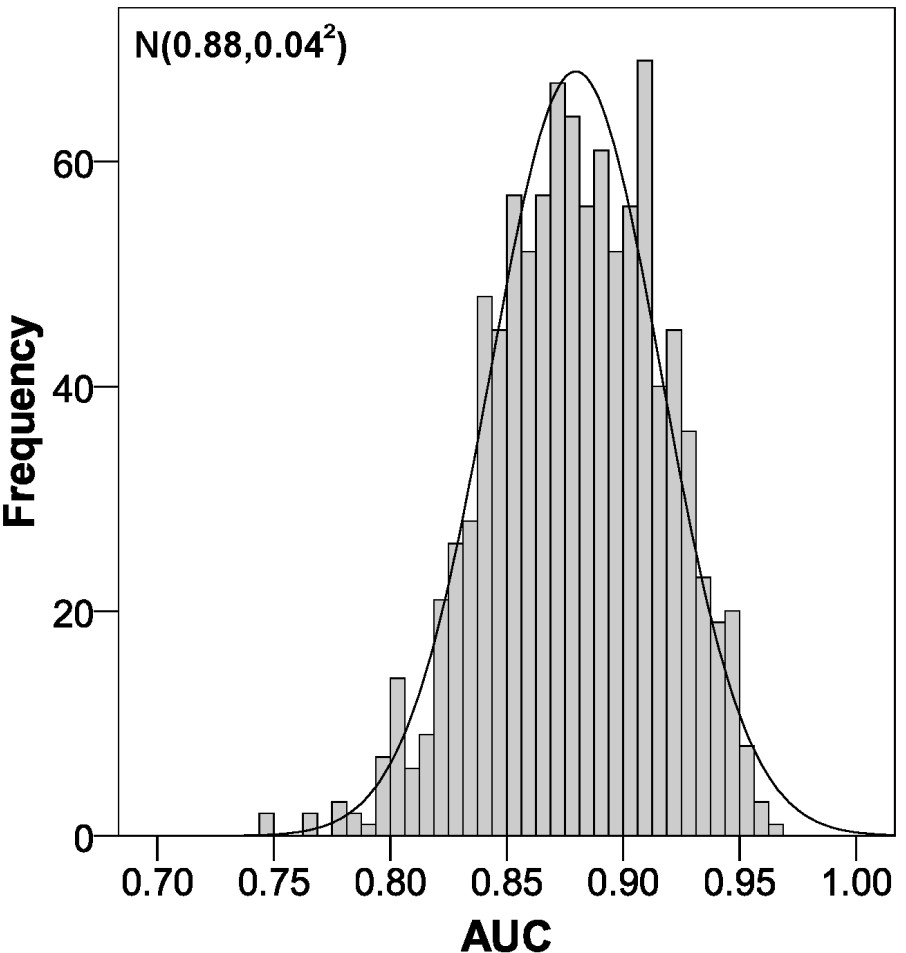

**Figure 2** **Area under the ROC curve distribution using bootstrapping to externally validate the proposed cut-off points for foveal thickness.** AUC, area under the ROC curve. Proposed cut-off points for foveal thickness: 90 μm in the presence of intraretinal fluid and 310 μm otherwise.

310 μm otherwise) (*Hernández-Martínez et al., 2015*). This validation was performed on 1,000 bootstrap samples, which gives greater accuracy to the results obtained (*Steyerberg et al., 2001*).

Similarly, we could say that the presence of intraretinal fluid confirms the diagnosis of macular oedema with almost any macular retinal thickness, whereas in order to confirm macular oedema conclusively in the absence of intraretinal fluid, a macular retinal thickness of 310 μm is necessary.

## Strengths and limitations of the study

The main strength of our work is the novel way in which the clinical question was addressed by externally validating the cut-off points for foveal thickness taking into consideration the presence of intraretinal fluid. This is the key point for being able to use a diagnostic test in routine clinical practice. In addition, this validation was not performed in just one sample

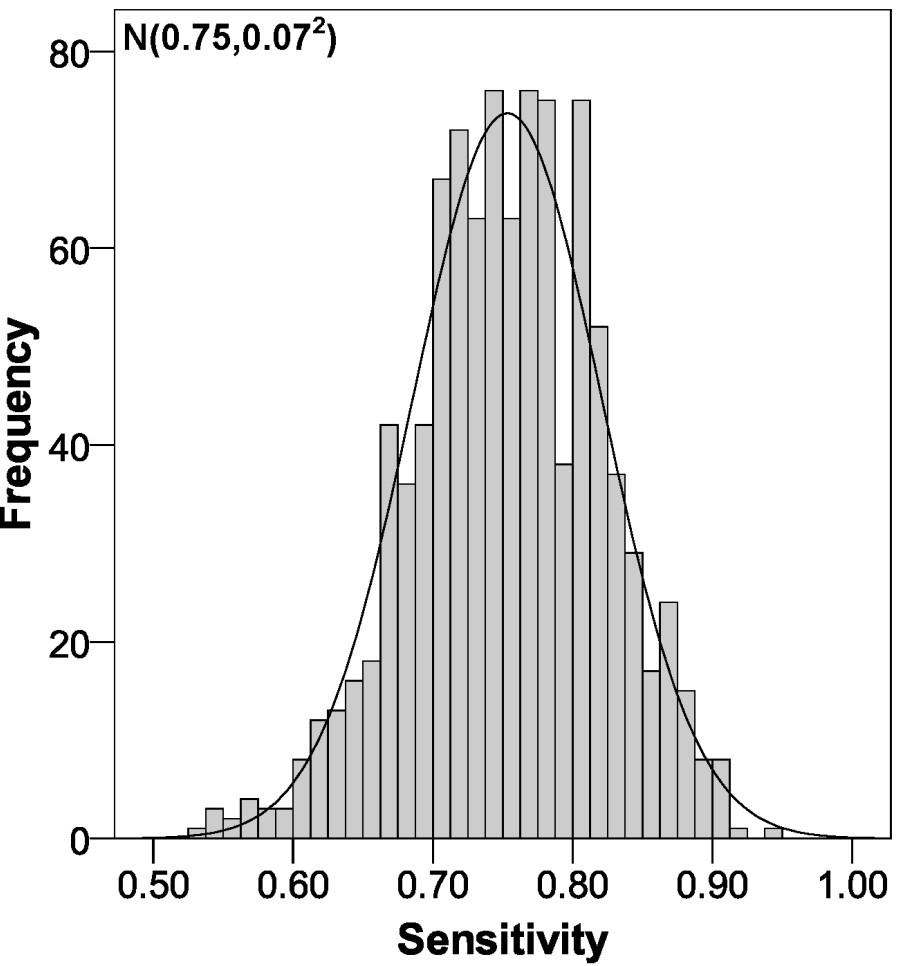

**Figure 3** **Sensitivity distribution using bootstrapping to externally validate the proposed cut-off points for foveal thickness.** Proposed cut-off points for foveal thickness: 90 μm in the presence of intraretinal fluid and 310 μm otherwise.

(split samples), rather it was corroborated in 1,000 bootstrap samples with very accurate and satisfactory results.

When examining the possible limitations of our study, one might consider that we introduced selection bias, since the characteristics of our patients are different from those of the original study (*Hernández-Martínez et al., 2015*). However, we must bear in mind that we are validating a diagnostic test through the calculation of sensitivity and specificity, independent parameters of disease prevalence (*Lalkhen & McCluskey, 2008*). In other words, the use of a sample obtained in specialised care consultations is not a selection bias. In addition, our sample was selected completely at random. Furthermore, all the fundamental variables of this study (diagnosis of diabetic macular oedema, foveal thickness and presence of intraretinal fluid) were measured by specialised professionals with validated devices. Finally, the precision for our sample size was around 6% based on the test specificity. Thus, validation studies involving larger sample sizes should be done

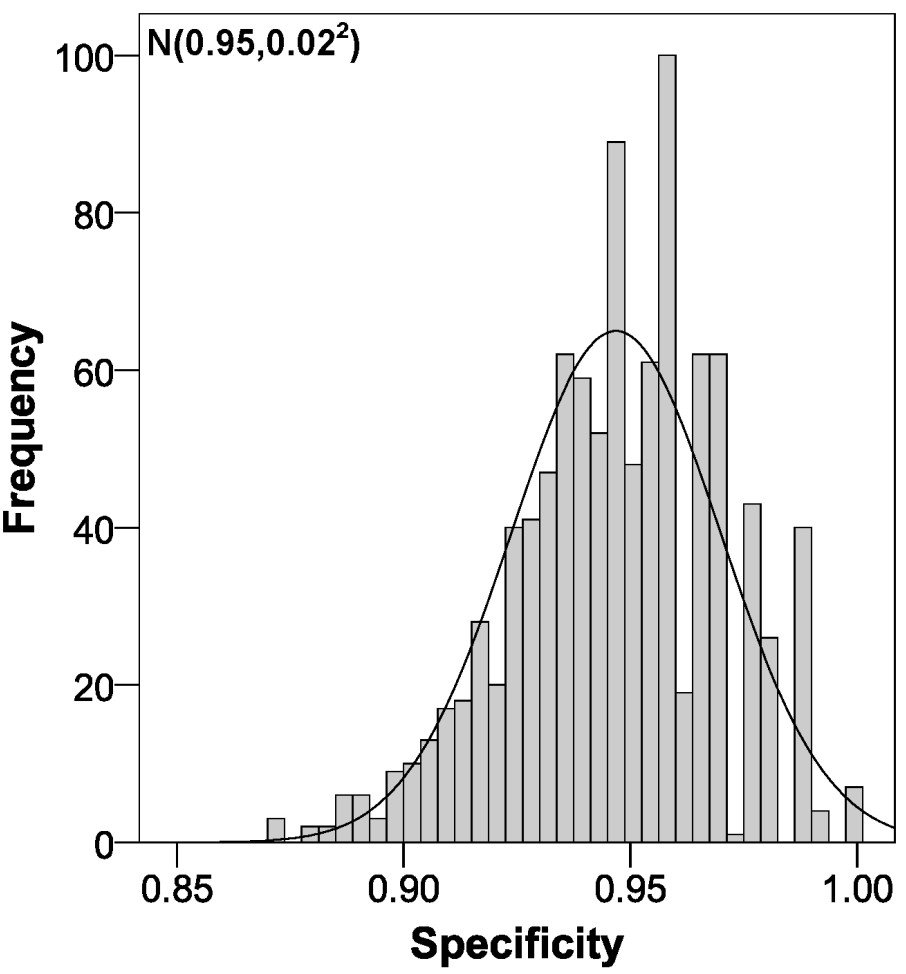

**Figure 4** **Specificity distribution using bootstrapping to externally validate the proposed cut-off points for foveal thickness.** Proposed cut-off points for foveal thickness: 90 μm in the presence of intraretinal fluid and 310 μm otherwise.

to reduce this error. Nonetheless, we performed the analysis with the bootstrap technique in order to obtain stable results (*Steyerberg et al., 2001*), using 1,000 random samples with replacement taken from the original sample.

## Comparison with the existing literature

In the absence of previous validation studies of the cut-off points indicated by the article published in late 2015 (*Hernández-Martínez et al., 2015*), we could only compare our results with those of the original study. In the AUC we obtained nearly the same value (0.88 versus 0.89), which indicates that the discrimination is practically the same. Very similar results were also obtained in terms of sensitivity and specificity, with an 8% decrease in sensitivity, while specificity increased by 6%. Clearly, as the likelihood ratios depend on sensitivity and specificity, our values varied accordingly, with the PLR increasing from 7.28 to 14.31 and the NLR increasing from 0.19 to 0.26 (*Hernández-Martínez et al., 2015*). Now, if we look at the PLR, when we obtain a value greater than 10, we can confirm the diagnosis

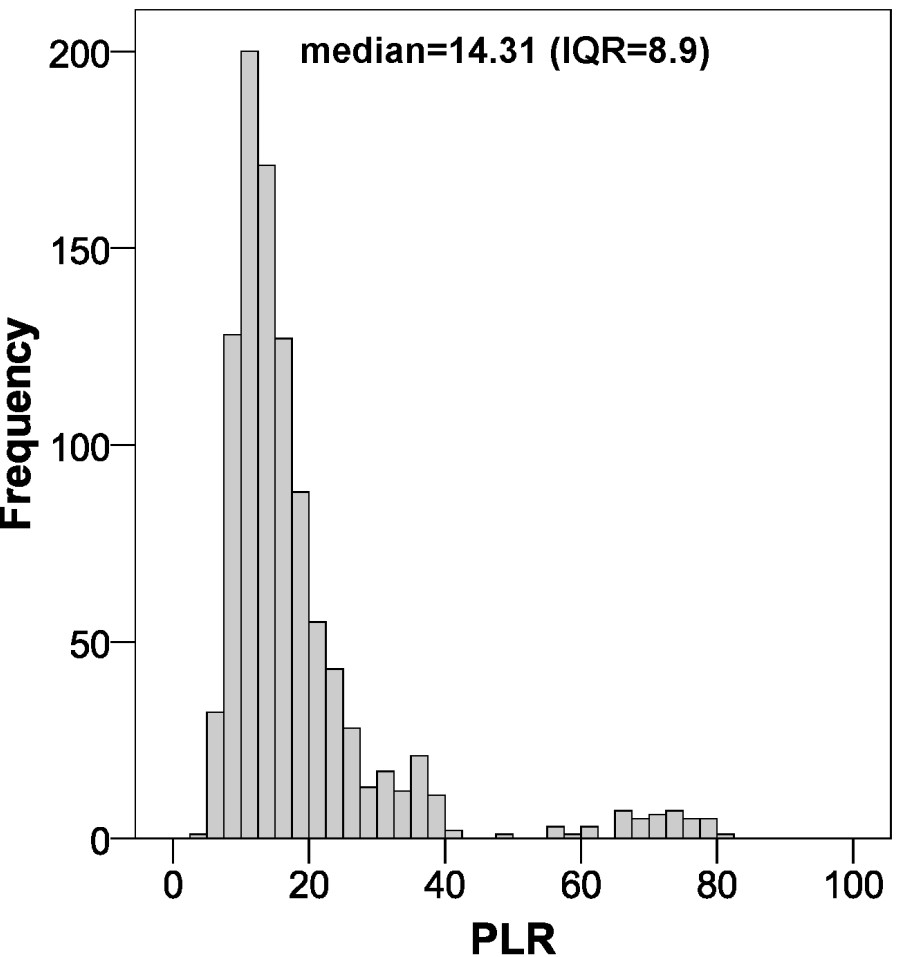

**Figure 5** **Positive likelihood ratio distribution using bootstrapping to externally validate the proposed cut-off points for foveal thickness.** PLR, positive likelihood ratio. Proposed cut-off points for foveal thickness: 90 μm in the presence of intraretinal fluid and 310 μm otherwise.

of macular oedema in an epidemiologically conclusive way, since it is 14.31 times more likely that the patient has the disease (*Guyatt & Drummond, 2002*).

## Implications for clinical practice and research

We have externally validated the cut-off points for foveal thickness to diagnose diabetic macular oedema resulting from the original study (90 μm in the presence of intraretinal fluid and 310 μm otherwise) in a population referred to ophthalmology services. The original study was validated in primary care (*Hernández-Martínez et al., 2015*). Thus we now have a decision algorithm to diagnose diabetic macular oedema in these two areas (primary and specialised care).

Given that the validation was performed in the health area covered by our hospital, we encourage other researchers to perform validation in their own communities. If similar results are found, these cut-off points for foveal thickness can be used in routine clinical practice (*Hernández-Martínez et al., 2015*).

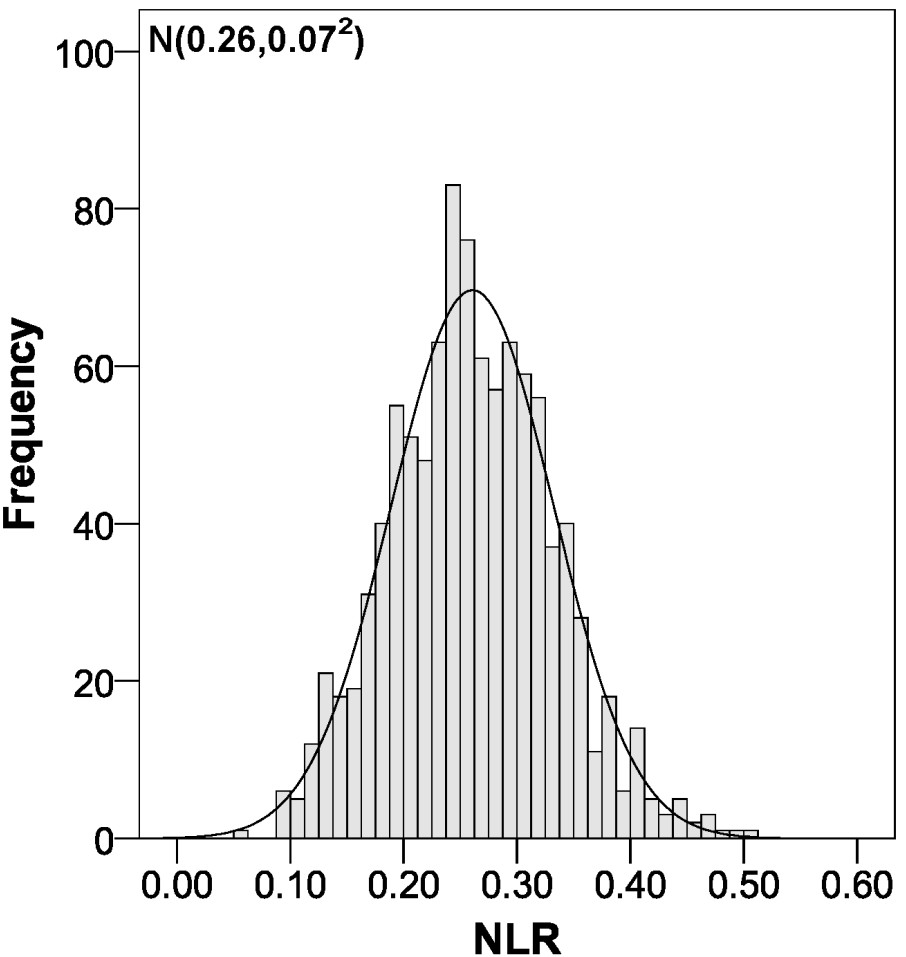

**Figure 6** **Negative likelihood ratio distribution using bootstrapping to externally validate the proposed cut-off points for foveal thickness.** NLR, negative likelihood ratio. Proposed cut-off points for foveal thickness: 90 μm in the presence of intraretinal fluid and 310 μm otherwise.

The results of our study have great clinical relevance in the screening for diabetic macular oedema, as we are validating a diagnostic test with a high discriminative capacity and good calibration (sensitivity and specificity). This results in a much reduced rate of false positives and false negatives for every 1,000 patients referred. Out of this total of 1,000 patients, 888 are diagnosed correctly. In addition, there are 37 patients whom we are diagnosing with diabetic macular oedema but who do not really have it and 75 patients who we consider not to have the condition but who do in fact have it. Considering that we are assessing a screening test for diabetic macular oedema, these error rates are low (<15%) and allow us to free up ophthalmology consultation offices as this test can be performed in primary care offices by health care personnel not necessarily specialised in ophthalmology. This is mainly because latest generation OCT devices focus the image of the retina automatically (*Hernández-Martínez et al., 2015*). In addition, the protocol dictates that the false negatives will return in a year for another screening test and at this time they will be detected, whilst

the false positives will have their initial diagnosis of diabetic macular oedema ruled out in the specialist ophthalmology service.

## CONCLUSION

The cut-off points for foveal thickness to diagnose diabetic macular oedema according to the presence of intraretinal fluid (90 µm in its presence and 310 µm otherwise) were externally validated through bootstrapping in a diabetic population referred to specialised ophthalmology services. For use in other communities, similar validation studies should be carried out.

## ACKNOWLEDGEMENTS

The authors thank Maria Repice and Ian Johnstone for help with the English language version of the text.

### Funding
The authors received no funding for this work.

### Competing Interests
Antonio Palazón-Bru is an Academic Editor for PeerJ.

### Author Contributions
- Carmen Hernández-Martínez conceived and designed the experiments, performed the experiments, wrote the paper, reviewed drafts of the paper.
- Antonio Palazón-Bru conceived and designed the experiments, analyzed the data, wrote the paper, prepared figures and/or tables, reviewed drafts of the paper.
- Cesar Azrak and Aída Navarro-Navarro conceived and designed the experiments, performed the experiments, reviewed drafts of the paper.
- Manuel Vicente Baeza-Díaz and José Juan Martínez-Toldos conceived and designed the experiments, performed the experiments, contributed reagents/materials/analysis tools, reviewed drafts of the paper.
- Vicente Francisco Gil-Guillén conceived and designed the experiments, contributed reagents/materials/analysis tools, reviewed drafts of the paper.

### Human Ethics
The following information was supplied relating to ethical approvals (i.e., approving body and any reference numbers):

All patients gave their written informed consent. The study was approved by the Ethics Committee of the General University Hospital of Elche and complies with the World Medical Association Declaration of Helsinki and with the standards described by the European Union Guidelines for Good Clinical Practice.

## Data Availability

The raw data has been provided as Data S1.

## Supplemental Information

Supplemental information for this article can be found online at http://dx.doi.org/10.7717/peerj.3922#supplemental-information.

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
