# Peer review of "External validation of cut-off points for foveal thickness taking into account the intraretinal fluid using optical coherence tomography to diagnose diabetic macular oedema"

_PeerJ, doi:10.7717/peerj.3922_

## Round 0.1 · original submission · Minor Revisions

· Academic Editor

Minor Revisions

The authors appealed the prior decision, and as a result their original submission was re-reviewed (by two new reviewers, #3 and #4).

In light of their feedback my decision is amended. Please provide a newly revised manuscript, taking into account the revisions you proposed in your appeal and the new (and prior) review comments.

· Appeal

Appeal

Dear Editorial Committee

The purpose of this letter to appeal against the decision concerning our manuscript #2017:01:15852 titled “External validation of cut-off points for foveal thickness taking into account the intraretinal fluid using optical coherence tomography to diagnose diabetic macular oedema”, which was rejected by the academic editor based on the comments of the reviewers of our paper. The editor mentions that the reason for rejection was that the study had methodological problems according to the opinion of Reviewer #1 (“Many concerns on experimental, and methodological issues have been expressed, in particular by one of the reviewers, that currently preclude further consideration of this manuscript”).

Most of the comments by this reviewer are not wholly correct, particularly those relating to the statistics and parts of the ophthalmology. Reviewer #2, on the other hand, states that the study is valid and only needs a couple of small changes (“The manuscript is well written and the scientific english language is adequate. References, figures and structure of the article are correct and meet the objective of the authors and the journal's recommendations. In the discussion section, I recommend to clarify the clinical benefit that can have an ophthalmologist reading this paper and to implement the introduction section analyzing in more detail the purpose of the article. Data and statistical analysis are correct, and the value of the research may have globally a good level of impact.”).

Accordingly, we respectfully request a re-evaluation of the manuscript (maybe by a younger more up-to-date specialist with a better command of statistics and the English language), especially bearing in mind that we have now acted on many of the comments (see our point-by-point answer). We are attaching this revised version, showing the changes made in comparison to the original version.

Thanking you for your consideration, we remain,

Dr Palazón-Bru (corresponding author and academic editor for PeerJ)


· · Academic Editor

Reject

Many concerns on experimental, and methodological issues have been expressed, in particular by one of the reviewers, that currently preclude further consideration of this manuscript.

Reviewer 1 ·

Basic reporting

.

Experimental design

.

Validity of the findings

.

Additional comments

1 pdf line 41 “ist (No authors listed, 1985; Kinyoun et al., 1989). Screening for diabetic retinopHY “
This is not a correct refeence. Why not use numbers as is now almost invariable in reference list and put the numbers in the text? The actual rerence ( pdf line261 is missing the year
2: lines38-41 grammar poor The reviewer re-writese:
….is indirect stereoscopic biomicroscopy using a (60)D lens.after mydriasis. ( ETDRS report –
3: line 43 ? Retinography?
4: line 46 “ due to “ implies a causal relationship. Has this really been established?
5: lines 49-50 “in the ophthalmological consultations. “ replace by a full stiop – cut it out
6: Line 54 “. In other words, the influence of intraretinal fluid has not been considered.” This is not logical and also it is not “ in other words”. If there is intraretinal fluid then either the retina is thicker or retinal cells must have vanished. I think this requires much more explanation. Much of what is referred to here is usually thought of as retinal cysts. Thickening without cyst formation would be an important variable
7: Foveal thickness. The foveal is a very small area, and the macula is much bigger. The authors should make a distinction between clinically non significant macular oedema and clinically significant macular oedema.. I think a figure illustrating OCT findings in the various categories they propose would make things clear
8: Line 60- I do not know what is meant here by bootstrapping. The area under the ROC curve is a method of obtaining specificity and sensitivity values It is not a calibration. Its setting a criterion. Is this the word the authors wanted? It permits one to obtain optimal values for dividing categories. These depend upon several factors and one is the proportion of patients who have the condition. Its no use having high specificity if very numerous patients have t be examined- the number of false positives wil be so large that the criterion cannot be used-the service will be overwhelmed. I think up to line 68 should be revised.
9: Line 72 The “who” implies that after “Elche” there should be the word “were”
10: Line 76 an area of 169555 inhabitants” Catchment area perhaps? Wrong power!
inhabitants number =x1 while area =x2
11: line 89 But early Alzheimer causes reduction in the nfl layer! The authors should check the liturarture.
12 :lines 91-96 I have already commented that a very similar sentence is poor English.
13: lines 105 107 The authors must understand “ an area of 1000 cetrnal microns “ is not normal scientific use. “valuation” is used incorrectly.
14: line 124 What does accuracy mean here? Given the strange way s quantitative data have been handled, the reviewer suggest that that the authors state the number of false positives and false negatives per 1000 referrals,
15: line 147 The statistical package may have worked but what is written is slightly nonsensical. Putting in decimal points when dealing with individuals is illlogical. See table1
Table 1 caption What does this mean?
“n(%), absolute frequency (relative frequency)”; Why not insert words into the table ? The revewer has used have used medical statistics for over 50 years, and have never seen S used as an indicator of standard deviation. If the figures given aree mean values, don’t we want to know the standard error of the mean? This is usually indicated by s.e. or s.e.m. The variables important to diagnosis include age and weight Surely these were recorded? And nephropathy? And crdiac nsfficiency ? Other interesting differences found in our work is country of origin. Do migrant workers form part of the sample?
16: line 169 The reviewer feels that given what the authors describe as intraretinal fluid is what he would call a cyst
17: The reviewer cannot comment on this bootstrap technique . Figs 1-5 are meaningless to him. He would like to see the ROC curves
18:line 181 “However, we must bear in mind that we are validating a diagnostic test through the calculation of sensitivity and specificity, independent parameters of disease prevalence (Lalkhen & McCluskey, 2008). In other words, the use of a sample obtained in specialised care consultations is not a selection bias “ The reviewer does not agree The optimal specificity and sensitivity of a test does depend upon the severity and frequency of a condition in the population which hs been recruited,
19: line 185 “ In addition, our sample was selected completely at random.” How can this assertion be made without providing evidence? Screening of populations is tricky. And the present sample of patients is so different clinically to the one provided by table 1 column 1 that small differences between optimal results is not surprising or even noteworthy.
20: for all the references If DOIs or other unique codings are not given (and they are not) then the year of publication must bre inserted.

Reviewer 2 ·

Basic reporting

No comment

Experimental design

No comment

Validity of the findings

No comment

Additional comments

The manuscript is well written and the scientific english language is adequate. References, figures and structure of the article are correct and meet the objective of the authors and the journal's recommendations. In the discussion section, I recommend to clarify the clinical benefit that can have an ophthalmologist reading this paper and to implement the introduction section analyzing in more detail the purpose of the article. Data and statistical analysis are correct, and the value of the research may have globally a good level of impact.

·

Basic reporting

Innovative

Experimental design

Well conceived

Validity of the findings

Good

Additional comments

Over all a worthwhile manyscript.
One major spelling error in abstract: Optimal Coherence Tomography should read as Optical Coherence Tomography

Reviewer 4 ·

Basic reporting

no comment

Experimental design

no comment

Validity of the findings

the sample size was 134 eyes. To estimate whether this sample was sufficient to validate the diagnostic test, the Authors computed the specificity estimation using the following parameters: an accuracy of 6.36% was obtained. However, since the sample is not big, I suggest discussing it as a potential, not strict, limit.
I agree with the conclusion that given that the validation was performed in the health area covered by their hospital, the Authors encourage other researchers to perform validation in their own communities and only if similar results are found, these cut-off points for foveal thickness can be used in routine clinical practice

Additional comments

This study contains interesting observations

---

## Round 0.2 · Major Revisions

· Academic Editor

Major Revisions

Your revised manuscript was sent back to Reviewer 1. They feel that some improvements are still necessary prior to publication. Please address each and every point raised by the Reviewer in a revised manuscript.

Reviewer 1 ·

Basic reporting

.

Experimental design

.

Validity of the findings

.

Additional comments

General
It seems to this reviewer that what the authors have done is to analyse their OCT results in their diabetic patients, and correlated these with clinical opinion derived from slit-lamp microscopy. From this analysis they have obtained values for OCT -derived measures of maximal retinal thickness which can occur in diabetics who do not have clinically significant diabetic macular oedema. The work seems well done. There are a number of mostly linguistic problems in the writing. There is no way of knowing whether these results are applicable to any other hospital or to other maker’s instruments. The MS refers to previous work (2015) published by the same group which seems to the reviewer very similar to the present MS. What is new is that the 2015 results are reproduced by the data of 2017

In detail
Title Would “independent” be better than External? Better word order is “ ….for foveal thickness using optical coherence tomography in the presence or absence of intraretinal fluid in patients with diabetic macular oedema”
Line 41 delete “by the ophthalmologist”
Line 41 Wrong method of citation Possibly “clinical manual Boston “ This is a matter for house style to decide
Line 49 Better word order would be “ Optical coherence tomography (OCT), which has been available for several years, allows us to perform a quantitative and qualitative study of diabetic macular oedema by providing a cross-sectional image of the retina, at high magnification, and automatically measures the thickness of various retinal layers. The OCT also shows the abnrormalities in structures such as the presence of cysts snd the accumulation of fluid “.
Line 53 “ two parameters” The reviewer does not understand what is meant. DMO most often begins extrafoveally, and the retina near the macula may contain cysts etc while the fovea is unaffected. Do the authors mean “ macula” where thy write “fovea”?
Line 54-57 “trying to determine a cut-off point from which we can say that there is diabetic macular oedema. In other words, the influence of intraretinal fluid in the OCT has not been considered, as when evaluating a diagnostic test it is possible that the presence of intraretinal fluid on OCT does not agree with the Gold standard” The reviewer dos not really understand what is meant. In assessing an individual, OCT is so much more powerful than slit lamp with a70D lens that there is no point raising this. We differentiate between sight-threatening and non-sight threatening macula oedema. Is this what the authors are attempting to discuss?
Line 62 The reviewer suggests that an illustration showing retinas with and without intraretinal fluid would make things plain!
Line 63-64 The fovea is the region where the OCT shows a pit, because there are only photoreceptors, and a thickness of the retina at this point of 90 microns is reasonable. But what is this 310 micron figure? Does this mean that if the fovea is not involved a thickness of 310 microns in extra-foveal retina indicates clinical significance? ( sight-threatening)
Line 66 Why use AUC for the Receiver Operating Charcteristic? Its almost invariably referred to as ROC for obvious reasons
Line 66 The sensitivity and specificity of a diagnostic test depends on the ROC but also on the values taken. This is in no sense a “calibration” The cut off points are arbitrary. If one wants to include almost all patients, ( that is, to have a high sensitivity ) then the specificity will decrease. The value finally adopted depends on various factors which include the prevalence of the condition, the rate of progress and the severity and also the resources available to track and treat the patients after the intial examination.
Line 67. The reviewer is going to accept the statement that there is only one publication giving statistical data about OCT measurements in patients with DMO , and he agrees that confirmatory data in different centres with different patient populations is required. Thus the results presented below are publishable. But a quick search of PUBMed gives 321 papers with some information about this subject ( even if not formalised ) since 2005!
Line 72 “ in our community” It is certain if you have a new instrument-of any sort- it is a good idea to collect your own normal values, even if the instrument has been well tested and comes with normative results. But such data are not necessarily for publication!
Line 88 The authors can determine whether their results are in agreement with those of Hernadez-Martinez : to call this ‘externally validating’ is a bit presumptious! What would the authors have done if their analysis did not agree with this previous paper?
Line 92 “ wanted to participate” -the reviewer rather worries about this. It is conceivable that this could result which was biased and was no generally applicable. .
Line 93 Ot is usual to say “exclusion criteria were : dementia. Recent cataract surgery…..”, This is a linguistic point- the conditions cannot be criteria ( in English)
Line 100 dilated pupil indirect ophthalmoscopy would read better
Line 103 These examinations were carried out by expert…
Line 104 Does this mean that the same doctors were used as in the previous cited publications?If so this whole project is scarcely independent! This is important!
Line 123 Was renal function measured?
Statistical methods as described seem standard, but they cannot produce what is referred to ss a calibration ( line 142)

Line 158 This data has been given previously see line 124 There are other such repetitions which shouldbe cleaned up.
Line 193 If the patient population equipment and operators were exactly the same as those in the 215 paper, then there would be no point in doing the work at all because it would not establish any generality: all that could be said would be that for these limited conditions th test was reliable.
Line 198 Earlier it was stated that randomness was not achieved.
Line 218 “externally validated” see reviewers previous comments
Line 223 This suggests to the reviewer that this MS deals with patients passing through the same hospital, seen by the same doctors, and examined with the same equipment, as in the 2015 project: and despite there being minor differences between this patient group and the one in the 2015 project, the results are very similar. Bujt what information in this paper would be of immediate use to a hospital in, for example, Venezuela?
Figures The reviwer suggests an additional figure with OCT images It would be nice to see an ROC graph. Are any of the other figures really required?

---

## Round 0.3 · accepted · Accept

· Academic Editor

Accept

The authors have satisfactorily answered to all the issues raised by the reviewers, including reviewer 1.